# Accelerating High b-Value Diffusion-Weighted MRI Using a Convolutional Recurrent Neural Network (CRNN-DWI)

**DOI:** 10.3390/bioengineering10070864

**Published:** 2023-07-21

**Authors:** Zheng Zhong, Kanghyun Ryu, Jonathan Mao, Kaibao Sun, Guangyu Dan, Shreyas S. Vasanawala, Xiaohong Joe Zhou

**Affiliations:** 1Departments of Radiology, Stanford University, Stanford, CA 94305, USA; 2Center for Magnetic Resonance Research, Chicago, IL 60612, USA; 3Henry M. Gunn High School, Palo Alto, CA 94306, USA; 4Department of Radiology, Neurosurgery and Biomedical Engineering, University of Illinois Chicago, Chicago, IL 60607, USA

**Keywords:** convolutional recurrent neural network, CRNN, diffusion MRI, DWI, non-Gaussian diffusion model, continuous-time random walk, CTRW, deep neural-network

## Abstract

Purpose: To develop a novel convolutional recurrent neural network (CRNN-DWI) and apply it to reconstruct a highly undersampled (up to six-fold) multi-b-value, multi-direction diffusion-weighted imaging (DWI) dataset. Methods: A deep neural network that combines a convolutional neural network (CNN) and recurrent neural network (RNN) was first developed by using a set of diffusion images as input. The network was then used to reconstruct a DWI dataset consisting of 14 b-values, each with three diffusion directions. For comparison, the dataset was also reconstructed with zero-padding and 3D-CNN. The experiments were performed with undersampling rates (R) of 4 and 6. Standard image quality metrics (SSIM and PSNR) were employed to provide quantitative assessments of the reconstructed image quality. Additionally, an advanced non-Gaussian diffusion model was employed to fit the reconstructed images from the different approaches, thereby generating a set of diffusion parameter maps. These diffusion parameter maps from the different approaches were then compared using SSIM as a metric. Results: Both the reconstructed diffusion images and diffusion parameter maps from CRNN-DWI were better than those from zero-padding or 3D-CNN. Specifically, the average SSIM and PSNR of CRNN-DWI were 0.750 ± 0.016 and 28.32 ± 0.69 (R = 4), and 0.675 ± 0.023 and 24.16 ± 0.77 (R = 6), respectively, both of which were substantially higher than those of zero-padding or 3D-CNN reconstructions. The diffusion parameter maps from CRNN-DWI also yielded higher SSIM values for R = 4 (>0.8) and for R = 6 (>0.7) than the other two approaches (for R = 4, <0.7, and for R = 6, <0.65). Conclusions: CRNN-DWI is a viable approach for reconstructing highly undersampled DWI data, providing opportunities to reduce the data acquisition burden.

## 1. Introduction

Diffusion-weighted magnetic resonance imaging (DW-MRI or DWI) can probe tissue microstructural alterations and has been increasingly used to study many disease processes, such as cerebral ischemia [1], brain tumors [2], focal liver lesions [3], breast cancer [4], and Parkinson’s disease [5]. The diffusion process of the water molecules in human tissue has been well demonstrated to be non-Gaussian, especially when probed under high b-values (e.g., >1500 s/mm^2^). To account for the non-Gaussian property of the diffusion process, a multi-b-value DWI approach together with a number of advanced diffusion models has been proposed [6,7,8,9,10]. Among the many non-Gaussian models, continuous-time random walk (CTRW) is of particular interest, as it provides two additional parameters, *α* and *β*, that are closely linked to tissue intra-voxel spatial and temporal heterogeneities. The CTRW model has been successfully applied to characterize Parkinson’s disease patients [10] and grade human brain tumors [11,12,13].

The data acquisition of multi-b-value DWI is typically performed with a fast MRI sequence, single-shot echo planar imaging (ssEPI), due to its rapid scan speed and resilience to motion [14]. Despite these merits, this technique is subject to geometric distortion arising from eddy currents and magnetic susceptibility variations among different tissues. Many approaches, such as multi-shot EPI [15] and the reduced field-of-view (rFOV) technique [16,17,18], have been proposed to address the issue by substantially shortening the echo train length of ssEPI. These approaches, however, often prolong acquisition times and/or accentuate the SAR issue [19]. An alternative strategy is to substantially undersample the k-space data, as in parallel imaging, compressive sensing (CS), and emerging deep learning techniques [20,21]. Compared with traditional parallel imaging techniques and CS-MRI, the deep learning approach can achieve a higher acceleration factor without compromising image quality.

Among the many deep learning neural networks, a convolutional recurrent neural network (CRNN) [22] is of particular interest because it combines the convolutional neural network (CNN) and the recurrent neural network (RNN), thereby providing better image quality by exploiting spatio-temporal redundancy in a series of images, such as the time series in dynamic imaging. Recognizing that the image series can be generalized to a set of diffusion images with different b-values and/or different diffusion directions, the CRNN approach can also be applied to reconstruct highly undersampled multi-b-value DWI data. 

Therefore, the goal of the present study was to employ a novel neural network—CRNN-DWI—and demonstrate its ability to achieve up to six-fold undersampling in DWI without degrading image quality. The reconstructed images were evaluated by comparing CTRW diffusion parameter maps obtained from the CRNN-DWI method versus the conventional approaches.

## 2. Materials and Methods

### 2.1. CRNN-DWI

Multi-b-value DWI series exhibited similar image features (i.e., edges, anatomy) among differing b-values and diffusion directions (Figure 1). CRNN-based network architectures merged the strengths of CNNs and RNNs, which made them particularly useful for spatial-temporal problems. Specifically, CNNs extracted common features from each weighted image, while RNNs aided in identifying patterns across varying b-values to reconstruct highly undersampled k-space data. The formulation of the proposed CRNN-DWI was expressed as:(1)$$X_{rec}=f_{N}\left(f_{N-1}\left({\ldots f}_{1}\left(X_{u}\right)\right)\right)$$
where $X_{rec}$ is the image to be reconstructed, $X_{u}$ is the input image series from a direct Fourier transform of the undersampled k-space data, $f_{i}$ is the network function including model parameters, such as the weights and biases of each iteration, and *N* is the number of iterations.

During each iteration, the network function $f_{i}$ performed:(2a)$$X_{rnn}^{\left(i\right)}=X_{rec}^{\left(i-1\right)}+CRNN\left(X_{rec}^{\left(i-1\right)}\right),$$
(2b)$$X_{rec}^{\left(i\right)}=DC\left(X_{rnn}^{\left(i\right)};X_{u}\right),$$
where CRNN is a learnable box consisting of five layers (Figure 2A), DC is a data consistency operation, and y is the acquired k-space data. The data-consistency operation refers to the requirement that the reconstructed image, when transformed back into the measurement (k-space), should agree with the acquired measurements. Specifically, we performed the hard data-consistency operation by replacing the acquired k-space data in the region where measurements were acquired [23].

Figure 2B shows the unfolded CRNN box, which consisted of one CRNN-***b***-i layer (evolving over both b-values and the iteration, green), three CRNN-i layers (evolving over the iteration only, blue), and one conventional CNN layer (yellow).

#### 2.1.1. CRNN-i Layer

For the CRNN-i layer (blue boxes), $H_{l}^{\left(i\right)}$ is the feature representation at layer l and iteration step i. In this layer, $W_{c}$ and $W_{r}$ represent the filters of input-to-hidden convolutions and hidden-to-hidden recurrent convolutions evolving over iterations, respectively, and $B_{l}$ denotes a bias term. We then had:(3)$$H_{l}^{\left(i\right)}=ReLU\left(W_{c}*H_{l-1}^{\left(i\right)}+{W_{r}*H}_{l}^{\left(i-1\right)}+B_{l}\right),$$
where ReLU is a rectifier linear unit, given by ReLU(x) = max(0, x). Here, $H_{l}^{\left(i\right)}$ had the shape of (batch size, T, 2, IMG_x_, IMG_y_), which was a representation of the entire *T* sequence. The convolution (*) was computed on the last two dimensions (IMG_x_, IMG_y_) of $H_{l}^{\left(i\right)}$.

#### 2.1.2. CRNN-b-i Layer

In this layer, both the iteration and the *b*-value information were propagated. Specifically, for each b in the *b*-value series, the feature representation $H_{l,b}^{\left(i\right)}$ was formulated as (Figure 2C):(4a)$$H_{l,b}^{\left(i\right)}=\vec{H_{l,b}^{\left(i\right)}}+\overset{\leftarrow }{H_{l,b}^{\left(i\right)}},$$
(4b)$$\vec{H_{l,b}^{\left(i\right)}}=ReLU\left(W_{c}*H_{l-1,b}^{\left(i\right)}+W_{b}*\vec{H_{l,b-1}^{\left(i\right)}}+{W_{r}*H}_{l,b}^{\left(i-1\right)}+\vec{B_{l}}\right),$$
(4c)$$\overset{\leftarrow }{H_{l,b}^{\left(i\right)}}=ReLU\left(W_{c}*H_{l-1,b}^{\left(i\right)}+W_{b}*\overset{\leftarrow }{H_{l,b+1}^{\left(i\right)}}+{W_{r}*H}_{l,b}^{\left(i-1\right)}+\overset{\leftarrow }{B_{l}}\right),$$
where $\vec{H_{l,b}^{\left(i\right)}}$ and $\overset{\leftarrow }{H_{l,b}^{\left(i\right)}}$ are the feature representations calculated along the forward and backward directions, respectively. Other parameters are defined in Figure 2C. 

### 2.2. Data Acquisition and Image Reconstruction

With IRB approval, multi-b-value DWI data were acquired from ten healthy subjects on a 3T GE MR750 scanner (GE HealthCare, Waukesha, WI, USA) using a single-shot EPI pulse sequence. The key acquisition parameters were: slice thickness = 5 mm, FOV = 22 cm × 22 cm, matrix = 256 × 256, slice number = 25, 14 b-values from 0 to 4000 s/mm^2^, and an acquisition time of ~6′30″. The acquired images were then transformed back to pseudo k-space and undersampled before being fed into the neural network. The undersampling mask pattern was a variable density pattern with the pseudo k-space center (24 lines) fully sampled. Seven datasets were used for training, two for validation, and one for testing. The datasets were also reconstructed with zero-padding and 3D-CNN for comparison. The experiment was performed with undersampling rates (R) of 4 and 6, respectively. The network was trained on a NVIDIA Titan Xp 64 GB graphics card. 

### 2.3. CTRW Model Fitting

Trace-weighted diffusion images were obtained by taking a geometric average from the original DWI images (used as ground truth) and reconstructed images from CRNN-DWI, 3D-CNN, and zero-padding, respectively. This was followed by fitting to a non-Gaussian CTRW model that could be described by the Mittag-Leffler equation:$$S/S_{0}=E_{\alpha }\left(-{\left(bD_{m}\right)}^{\beta }\right)$$
where *S* is the signal intensity of the trace-weighted diffusion images, *S*_0_ is the signal intensity of *b* = 0, α and β represent temporal and spatial diffusion heterogeneities, and *D_m_* is an anomalous diffusion coefficient. The equation was fit to the multi-*b*-value dataset and was run iteratively using a Levenberg–Marquardt method across the pixels of all 25 slices to generate parameter maps for each of the variables (α, β, and *D_m_*). The fitting process consisted of two major steps: (i) *D_m_* was estimated using a mono-exponential model with all *b*-values; and (ii) with the estimated *D_m_* from the previous step, *α* and *β* were simultaneously determined through a non-linear fitting. The fitting process was terminated after convergence was achieved, with a maximum of 80 iterations.

### 2.4. Image and Statistical Analysis

To assess the quality of the reconstructed images, standard indices, including structural similarity (SSIM) and peak signal-to-noise ratio (PSNR), were calculated and compared for the reconstructed images from the three different approaches (zero-padding, 3D-CNN, and CRNN-DWI). The qualities of the diffusion parameter maps (*α*, *β*, and *D_m_*) were also evaluated by calculating the SSIM values against the diffusion parameter maps from the ground truth for all slices. The SSIM values of each diffusion parameter were then grouped together and compared using a paired t-test among the three different reconstruction approaches. A *p*-value < 0.05 (after Bonferroni correction) was considered significant. All the comparisons were performed using Matlab (MathWorks, Inc., Natick, MA, USA).

## 3. Results

### 3.1. Reconstructed Images

Figure 3 and Figure 4 show a set of individual diffusion images (b = 1000 s/mm^2^ and b = 4000 s/mm^2^) using CRNN-DWI, with R = 4 and 6, respectively. The images reconstructed using CRNN-DWI had better image quality than the zero-padding and 3D-CNN approaches. This was evidenced by the higher SSIM and PSNR values. Specifically, the average SSIM and PSNR of CRNN-DWI were 0.750 ± 0.016 and 28.32 ± 0.69 (R = 4), and 0.675 ± 0.023 and 24.16 ± 0.77 (R = 6), respectively, both of which were much higher than those using zero-padding (SSIM/PSNR = 0.516/12.18 for R = 4; SSIM/PSNR = 0.479/12.04 for R = 6) or 3D-CNN reconstruction (SSIM/PSNR = 0.535/13.37 for R = 4; SSIM/PSNR = 0.505/13.84 for R = 6). The undersampling mask pattern for the acceleration factors of 4 and 6 are shown in Figure 3 and Figure 4, respectively.

### 3.2. Diffusion Parameter Maps

The representative CTRW parameter maps are shown in Figure 5. The parameter maps from CRNN-DWI were visibly less noisy and exhibited less artifacts than the parameter maps from the other two approaches. Quantitatively, the SSIM values from CRNN-DWI were also significantly larger than the other two approaches for both R = 4 and R = 6 (*p* < 0.01), as summarized in Table 1. Similar to the individual diffusion images, the SSIM values were also substantially improved in the CTRW parameter maps (for R = 4 larger than 0.8 and for R = 6 larger than 0.7) when CRNN-DWI was employed. The trace-weighted images and the signal decay curves from two randomly selected regions of interest agreed well between the images from CRNN-DWI (R = 4 and 6) and those from fully sampled data (Appendix A).

## 4. Discussion

A novel neural network—CRNN-DWI—was successfully applied to the reconstruction of a highly undersampled multi-b-value DWI dataset. With an up to six-fold reduction in raw data, CRNN-DWI worked well without noticeably compromising image quality or diffusion signal quantification. The images reconstructed from CRNN-DWI also performed well when analyzed with an advanced diffusion model (CTRW), yielding high SSIM values in the diffusion parameter maps when compared to the fully sampled dataset.

The CRNN approach was successfully applied to reconstruct highly under-sampled dynamic MRI datasets [22,24,25], where redundancy within temporal series was utilized. Similarly, the redundant image features among different b-values and diffusion directions allowed CRNN-DWI to exploit correlations within the dataset. The same approach could be extended to a larger number of b-values and/or diffusion directions (e.g., >60, as in a typical HARDI dataset). Note that we trained the neural network with only 175 images. The performance of the neural network could be further improved with more datasets to finetune the network. 

When applied to k-space undersampling, CRNN-DWI can potentially reduce image distortion associated with ssEPI sequences. Such distortion is proportional to the bandwidth of the EPI phase-encoding direction. The CRNN-DWI approach can also help reduce acquisition times when applied to other non-single-shot-based sequences, such as multi-shot EPI or fast spin-echo (FSE) [26]. In that scenario, the acquisition time is related to the k-space lines acquired per shot and the total number of k-space lines required for image reconstruction. Undersampling k-space means acquiring fewer k-space lines, which may even reduce multiple shots into one shot, thereby substantially reducing acquisition times.

Advanced diffusion models with multi-b-value diffusion-weighted images have been increasingly used in probing micro-structures of human tissues [8]. The CTRW model focuses on exploring the tissue heterogeneities by offering three quantitative parameters: *D_m_*, *α*, and *β*. Specifically, *D_m_* describes how fast the diffusion decays analogous to the conventional diffusion coefficient, whereas *α* and *β* are closely linked to temporal and spatial heterogeneities, i.e., the amount of time for water molecules to make a jump (temporal) and the displacement when making a move (spatial). As a result, these parameters can reflect different aspects of the diffusion heterogeneities in the tissue. This approach has already been successfully applied in assessing glioma, pediatric brain tumors, gastrointestinal cancer, bladder cancer, gastric cancer, prostate cancer, and Parkinson’s disease [2,10,11,27,28,29,30,31]. The result from this work implies that CRNN-DWI together with the CTRW model can be beneficial in exploring these and other applications.

The CRNN-DWI approach can be potentially extended to other advanced diffusion models. However, caution should be exercised, as advanced DWI measurements can be influenced by various factors, such as b-value, diffusion times, spatial resolution, magnetic field strength, etc. These factors can play a significant role in accurately constructing the diffusion models. Furthermore, the spatial distribution should also be taken into account, particularly for DWI-based measurements with large b-values and multiple diffusion directions, such as DTI and HARDI. Neglecting the spatial distribution may result in significant systematic errors [32]. To address this issue, one potential strategy is to conduct phantom studies incorporating b-matrix spatial distribution (BSD). After evaluating the non-uniformity caused by various scanners and imaging parameters, data correction can be performed to mitigate the concerns [32,33]. This approach is expected to help improve the accuracy and reliability of the results obtained from the diffusion models used in the studies.

One limitation of CRNN-DWI is the extensive GPU memory required in comparison with other neural networks. This is due to the large number of parameters that must be stored during the training process. Utilizing a deep subspace-based network may help mitigate this problem by reconstructing a simpler set of basic functions [34]. Another limitation of this study is that the performance of the proposed approach has not been evaluated using patient images that may include pathology and other abnormalities. This important aspect can be addressed in future studies, where the impact of the proposed method on disease diagnosis can be assessed.

## 5. Conclusions

To conclude, the CRNN-DWI is a viable approach for reconstructing highly undersampled DWI data, providing opportunities to reduce the data acquisition burden.

## Figures and Tables

**Figure 1 bioengineering-10-00864-f001:**
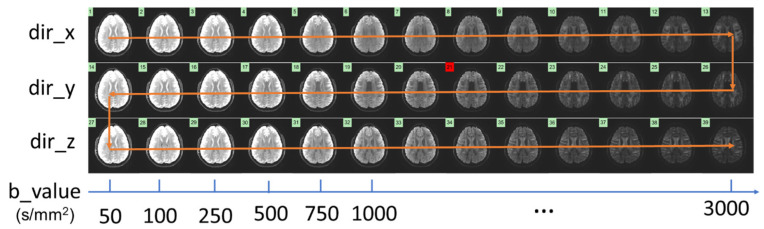
A set of multi-b-value DWI datasets from a representative slice showing the high degree of data redundancy among the images with different b-values and different diffusion directions. (The b-value ranged from 50 to 3000 s/mm^2^.)

**Figure 2 bioengineering-10-00864-f002:**
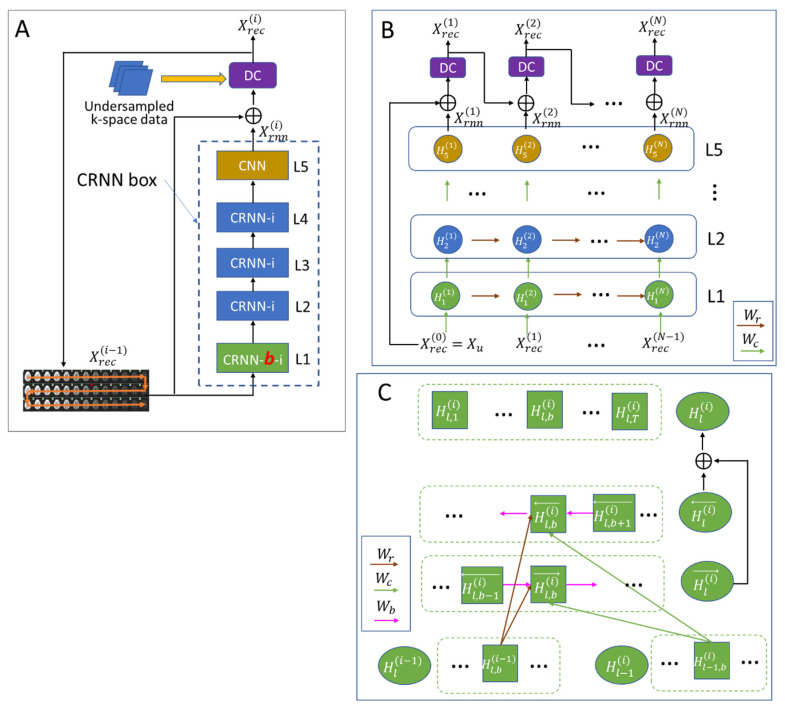
The structure of CRNN-DWI used in this study. (**A**) The detailed structure of the proposed network for each layer; (**B**) the unfolded structure of the proposed network for each iteration; and (**C**) the detailed structure of the CRNN-b-i layer. The green arrow ($W_{c}$), brown arrow ($W_{r}$), and pink arrow ($W_{b}$) represent the filters of input-to-hidden convolutions, hidden-to-hidden recurrent convolutions evolving over iterations, and the *b*-value series, respectively.

**Figure 3 bioengineering-10-00864-f003:**
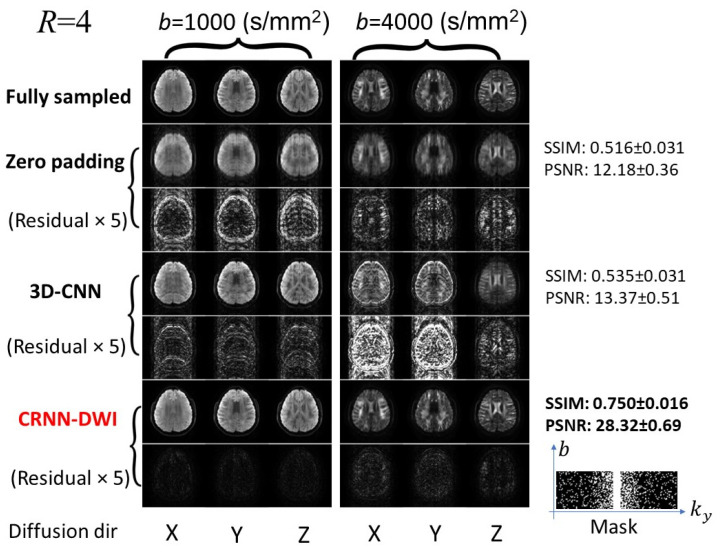
Representative images from the experiments with four-fold undersampling. The reconstructed images using CRNN-DWI outperformed zero-padding or 3D-CNN and were the closest to the ground truth for a broad range of b-values.

**Figure 4 bioengineering-10-00864-f004:**
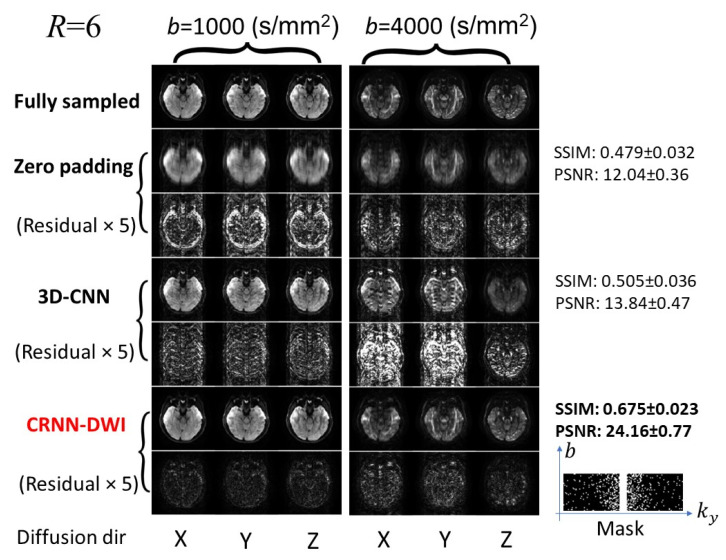
Representative images from the experiment with six-fold undersampling. Although the quality of the reconstructed images using CRNN-DWI was not as good as that of four-fold undersampling, the CRNN-DWI image quality still considerably out-performed that of zero-padding or 3D CNN.

**Figure 5 bioengineering-10-00864-f005:**
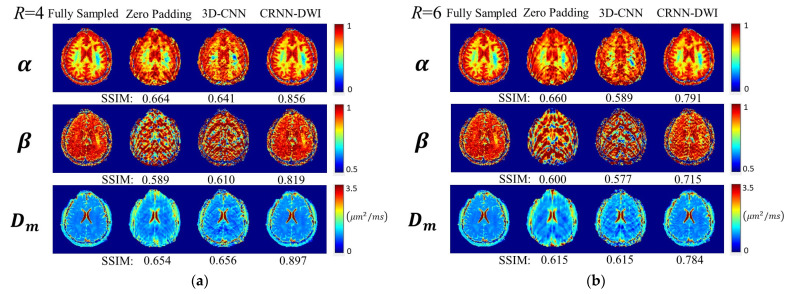
Representative CTRW parameter maps (*α*, *β*, and *D_m_*) calculated using the trace-weighted images from different reconstruction approaches for R = 4 (**a**) and R = 6 (**b**), respectively. For both acceleration factors, the parameter maps generated by the CRNN-DWI method showed significantly higher SSIM values when compared to those obtained from zero padding and 3D-CNN appraoches.

**Table 1 bioengineering-10-00864-t001:** The SSIM values of the diffusion parameter maps using different reconstruction approaches.

	CRNN-DWI			3D-CNN			Zero-Filling		
	*α*	*β*	*D_m_*	*α*	*β*	*D_m_*	*α*	*β*	*D_m_*
R = 4	0.84 ± 0.04	0.82 ± 0.04	0.9 ± 0.03	0.62 ± 0.06	0.62 ± 0.05	0.64 ± 0.06	0.66 ± 0.06	0.62 ± 0.05	0.65 ± 0.06
R = 6	0.75 ± 0.06	0.71 ± 0.05	0.77 ± 0.05	0.6 ± 0.06	0.6 ± 0.05	0.61 ± 0.06	0.64 ± 0.06	0.61 ± 0.05	0.62 ± 0.06

## Data Availability

Data used in this study will be available upon request.

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
