# Peer review of "Accelerating High b-Value Diffusion-Weighted MRI Using a Convolutional Recurrent Neural Network (CRNN-DWI)"

_bioengineering, 2023, doi:10.3390/bioengineering10070864_

Round 1

Reviewer 1 Report

Quite well written, interesting study on the use of CNN algorithms to improve DWI-based techniques. However, several significant changes and extensions are necessary.

If we want to teach CNN algorithms, we really need "groud truth" data. Do we have them here? And what should be improved?

1. It should be noted that the techniques listed (either deterministic or probabilistic) are based on DWI measurements. What is theoretically missing here?

In order to precisely determine the values of diffusion coefficients, diffusion coefficient, diffusion tensor, tensor kurtosis, etc. we need precise DWI measurements. To do this, we need a precise determination of the matrix b, which is a 3x3 matrix, symmetric, different for each diffusion gradient vector (DWI direction). Typically, there is a single matrix for each DWI provided by the manufacturer, potentially leading to large biases. And as recent works show, it actually has a spatial distribution.

Therefore, I am asking you to supplement this information and literature, I propose an analysis of the following issues:

- b-matrix,

- b-matrix spatial distribution,

- Generalized ST equation for non-uniform gradients,

- b-matrix spatial distribution in DWI/DTI

2. Another important issue is the discussion of the consequences of typical measurement approaches, i.e. not using the full b matrix or its actual spatial distribution.

As shown by theoretical works, numerical simulations, methodical experimental works on phantoms, single patients, in fact we have a spatial b-matrix distribution. Failure to take this into account leads to large systematic errors of a spatial nature. These errors cannot be removed by additional accumulations or by increasing the number of DWI directions.

This seems to be an important direction to improve techniques based on DWI. The topics for discussion here are:

- Systematic Errors in DWI/DTI

- Correction of Errors in DWI/DTI

- Validation of BSD-DTI

- Phantoms for BSD-DTI

- Anisotropic phantoms for BSD-DTI

In general, BSD-DTI seems to be the most comprehensive description of this problem at present.

This undoubtedly also has an impact on the any measured object, by DTI metrics or tractography.

3.

Another quite important issue is the variety of measurement parameters, value of b, resolution, times TE, RT, field B, generally the parameters of a single DWI measurement. These are quantities that affect from a physical point of view how we "see" diffusion coefficients from a physical point of view. This is also an important issue and requires a certain consensus in the future as well as additional information from MRI scanner manufacturers, for example about diffusion times, small and large delta. This can be analyzed in the context of the issue:

- NMR Diffusometry and Cellular System

The article is an attempt to model non-Gaussian diffusion. But which approach best reflects reality, how many components of the diffusion coefficient or tensor should we take into account? There is no consensus here, there are trials. What we can do is provide the best measured image without systematic errors. Then we can build models.

Still on the margin. How can you go back to k-space from the image, which is magnitude. It is impossible. The raw signal consists of re and im, this is subjected to FFT and from that we build a picture of the proton density which is the amplitude. There is no phase information anymore.

Unless you have it?

These are a few important points whose discussion, substantive and literature supplementation will enrich this undoubtedly promising manuscript.

quite well

Reviewer 2 Report

The novelty of the paper should be mentioned.

The title must be incorporated as per the existing proposed technology.

Equations must be typed in equation editors.

Innovation strategy must be incorporated as per the current 17 SDG goals

Do you think the proposed work will work in same domain if they consider innovation and other fields?

Can you define the range of Figure 1. A set of multi-b-value DWI datasets from a representative slice.

How many modalities may be considered for diagnosis analysis

Author should cite some of recent work on his area.

I strongly suggest a significant re-articulation of the entire draft for improving the readability.

Mentions and explain the CNN layer with respect of any disease.

Mention the order of the following heading   CRNN-i layer: 

Data Acquisition and Image Reconstruction: describe the RGB model.

Minor corrections in grammar or spelling mistakes.

Formatting must be considered to the relevant diagrams and paragraphs.

Round 2

Reviewer 1 Report

*R1.1 •    In order to precisely determine the values of diffusion coefficients, diffusion coefficient, diffusion tensor, tensor kurtosis, etc. we need precise DWI measurements. To do this, we need a precise determination of the matrix b, which is a 3x3 matrix, symmetric, different for each diffusion gradient vector (DWI direction). Typically, there is a single matrix for each DWI provided by the manufacturer, potentially leading to large biases. And as recent works show, it actually has a spatial distribution. Therefore, I am asking you to supplement this information and literature. As shown by theoretical works, numerical simulations, methodical experimental works on phantoms, single patients, in fact we have a spatial b-matrix distribution. Failure to take this into account leads to large systematic errors of a spatial nature. These errors cannot be removed by additional accumulations or by increasing the number of DWI directions. This seems to be an important direction to improve techniques based on DWI.

Response: We agree with the Reviewer that DWI (especially DTI) has spatial distribution. However, in this study, the diffusion model employed (CTRW model) was calculated based on the trace-weighted image, which involves taking the geometric average of three images with different diffusion directions. Consequently, the spatial distribution was averaged across the three diffusion directions. While DTI encompasses a greater number of diffusion directions, its analysis relies more heavily on the b-matrix and the associated spatial distributions. It is worth noting that the technique proposed in this work has the potential to be expanded to various avenues, including DTI. As a result, we have incorporated additional content into the revised manuscript to elaborate on the spatial distribution properties relevant to DTI (Page 7).

AD. Trace-weighted images are trace-weighted, which is obtained as ADC from 3 orthogonal DWI measurements (most common) or from e.g. DTI. Here we have the first case. That is, 3 DWIs, each burdened with an unknown systematic error (description from 1 review).

This should be emphasized.

In 2003, we note Bammer's observation of the connection of this fact with gradient coils. However, only the BSD-DTI approach sheds more light on the issue and shows that it depends on the MR sequence and its parameters, the MR scanner as well as the position in the laboratory setup (magnet and gradient coils), as well as shows an independent method for precise determination of the spatial distribution matrix-b based on the phantoms being the source of the standard diffusion tensor.

In general, BSD-DTI seems to be the most comprehensive description of this problem at present.

This undoubtedly also has an impact on the any measured object, by DWI based techniques, especially for large b.

In order to increase the precision of research and inference, it is good to show the improvement path. In reference 33 this is described a bit, but it would be nice to show a path for improvement. 1- Phantoms as diffusion tensor patterns, 2-BSD usage. 3- Evaluation of non-uniformity of magnetic field gradients (for a given sequence, parameters, scanner...)4 - Data correction.

*R1.1 •    In order to precisely determine the values of diffusion coefficients, diffusion coefficient, diffusion tensor, tensor kurtosis, etc. we need precise DWI measurements. To do this, we need a precise determination of the matrix b, which is a 3x3 matrix, symmetric, different for each diffusion gradient vector (DWI direction). Typically, there is a single matrix for each DWI provided by the manufacturer, potentially leading to large biases. And as recent works show, it actually has a spatial distribution. Therefore, I am asking you to supplement this information and literature. As shown by theoretical works, numerical simulations, methodical experimental works on phantoms, single patients, in fact we have a spatial b-matrix distribution. Failure to take this into account leads to large systematic errors of a spatial nature. These errors cannot be removed by additional accumulations or by increasing the number of DWI directions. This seems to be an important direction to improve techniques based on DWI.

Response: We agree with the Reviewer that DWI (especially DTI) has spatial distribution. However, in this study, the diffusion model employed (CTRW model) was calculated based on the trace-weighted image, which involves taking the geometric average of three images with different diffusion directions. Consequently, the spatial distribution was averaged across the three diffusion directions. While DTI encompasses a greater number of diffusion directions, its analysis relies more heavily on the b-matrix and the associated spatial distributions. It is worth noting that the technique proposed in this work has the potential to be expanded to various avenues, including DTI. As a result, we have incorporated additional content into the revised manuscript to elaborate on the spatial distribution properties relevant to DTI (Page 7).

Ad. 2 Ok.  You can give current digressions about models of the diffusion tensor, for example the number of components. But generally ok.

*R1.3 •    How can you go back to k-space from the image, which is magnitude. It is impossible. The raw signal consists of re and im, this is subjected to FFT and from that we build a picture of the proton density which is the amplitude. There is no phase information anymore.

Response: The k-space was generated by directly Fourier-transforming the diffusion images. We acknowledge the Reviewer's suggestion that using the original raw data, which preserves the phase information, would offer a more accurate representation of the actual acquisition. Unfortunately, the original raw k-space data was not available. Hence, the phase information was lost. We recognize the limitation of our approach. However, such limitation is not expected to significantly impact the overall conclusions of our study. This is because most deep neural networks treat the real and imaginary part separately when feeding the data into the neural network. Additionally, it is worth noting that a great number of deep learning studies (1–3) have also employed a similar approach, namely, performing Fourier transformation on the magnitude image to obtain the k-space data and conducting subsequent analysis without the inclusion of phase information.

Ad 3. This is an approximation, pseudo k-space, and basically a new model, and the term k-space should not be used. Phase information is irretrievably lost. From the module you cannot reconstruct re and im, which were experimentally obtained unambiguously.

ok

Author Response

*R1.1 •    In order to increase the precision of research and inference, it is good to show the improvement path. In reference 33 this is described a bit, but it would be nice to show a path for improvement. 1- Phantoms as diffusion tensor patterns, 2-BSD usage. 3- Evaluation of non-uniformity of magnetic field gradients (for a given sequence, parameters, scanner...) 4 - Data correction.

Response: We have taken the suggestion from the Reviewer, and added more information describing the path for further improvements.

*R1.2 •    This is an approximation, pseudo k-space, and basically a new model, and the term k-space should not be used. Phase information is irretrievably lost. From the module you cannot reconstruct re and im, which were experimentally obtained unambiguously.

Response: We thank the Reviewer for pointing this out and agree with the Reviewer regarding the phase. The term “k-space” was changed to “pseudo k-space” in the revised manuscript to accurately reflect the nature of the data used. Although the data may not represent actual k-space, the conceptual framework and analysis related to k-space remain valid. Therefore, this change does not impact the conclusions drawn from our study.

Reviewer 2 Report

All content must be improved and relevant.

All citations must be relevant.

Citation included 2023-2024

Accepted in present form.

All content must be improved and relevant.

All citations must be relevant.

Citation included 2023-2024

Accepted in present form.

Author Response

We thank the Reviewer for accepting the manuscript in its current form. If the Reviewer has any additional questions or requires further clarification, we would be more than happy to provide additional information to address his/her concerns.